# Geographic Mosaic of Extensive Genetic Variations in Subterranean Mole Voles *Ellobius alaicus* as a Consequence of Habitat Fragmentation and Hybridization

**DOI:** 10.3390/life12050728

**Published:** 2022-05-13

**Authors:** Valentina Tambovtseva, Irina Bakloushinskaya, Sergey Matveevsky, Aleksey Bogdanov

**Affiliations:** 1Koltzov Institute of Developmental Biology, Russian Academy of Sciences, 119334 Moscow, Russia; lynx1994@gmail.com (V.T.); bogdalst@yahoo.com (A.B.); 2Vavilov Institute of General Genetics, Russian Academy of Sciences, 119991 Moscow, Russia; sergey8585@mail.ru

**Keywords:** speciation, hybridization, chromosomal evolution, mito-nuclear discordance, Robertsonian translocations, *Ellobius*, subterranean rodents

## Abstract

Restricted mobility, sociality, and high inbreeding—characteristic for subterranean mammals—lead to rapid changes in their genome structure. Up to now, the Alay mole vole *Ellobius alaicus* was a data-deficient species; its spatial and phylogenetic relationships with a sibling species, *E. tancrei*, were not clarified. We carried out a genetic analysis including differential G-banding of chromosomes and mitochondrial (*cytb*) and nuclear gene (*XIST* and *IRBP*) sequencing. The phylogenetic reconstruction based on *cytb* represented the expected phylogenetic relationships of two species. Using the *XIST*, we revealed two new lineages among *E. alaicus* from the Alay Valley (Southern Kyrgyzstan). Analysis of *IRBP* demonstrated presence of the specific genotype in most of *E. alaicus* specimens, but also revealed the haplotype, typical for *E. tancrei*, in some Alay mole voles. The results may be explained as persistence of ancestral gene polymorphism in *E. alaicus* or limited interspecific hybridization with *E. tancrei*. Several chromosomal forms were revealed in *E. alaicus* in the Alay Valley. We propose that ‘mosaic’ genetic polymorphism might appear in *E. alaicus* due to fragmentation of their habitats in highlands of the Alay Valley, Tien Shan, and Pamir-Alay as well as due to hybridization with *E. tancrei* or persistence of ancestral polymorphisms.

## 1. Introduction

Subterranean animals are characterized by a number of significant peculiarities. Convergent similarity of morphological and physiological features originated due to digging activity and living underground in burrows [1,2]. Moreover, extensive intraspecific chromosomal variability has been described for many subterranean rodents [3,4]. The most chromosomally diverse genera (such as *Ctenomys, Oryzomys, Fukomys, Nannospalax, Ellobius*, etc.) are highly specialized; underground habitations limit their dispersal and shape foraging, mating, and breeding patterns [5,6,7]. The ranges of subterranean species are commonly represented as systems of isolated populations, which are characterized by high variability in the spatial genetic structure, small effective population sizes, low dispersal rates, high inbreeding, recurrent “bottleneck” events, limitations of genetic drift, etc. [8,9]. Chromosomal speciation has been suggested to occur in small demes, which are characterized by a succession of isolation and secondary contacts with neighboring groups of demes [10,11]. The strict environmental conditions of underground habitation and small effective population size can act as antagonistic factors, which are manifested in different rates of changes in the studied traits [12].

In the mountains, the compartmentalization of habitats (i.e., the mosaic of external conditions) may result in mosaic changes in the genome. Rodents are convenient for studying both chromosomal and molecular polymorphism—an evolutionarily young group with rapid alternation of generations, high population size, reproduction rate, and ecological lability. Species whose habitats are characterized by a significant variety of landscapes, climatic conditions, and other environmental factors are of primary interest. We consider the mole voles of the genus *Ellobius* (subfamily Arvicolinae) as a promising subject for studying this evolutionary problem and related mechanisms. Three sibling species of the subgenus *Ellobius*—*E. talpinus*, *E. tancrei*, and *E. alaicus*—show interesting differences in evolutionary pathways, including a unique sex chromosome system (XX in males and females). No chromosomal variability is known for *E. talpinus* s. str. (2n = NF = 54) [13,14], while for *E. tancrei*, high variability (2n = 54–30, NF = 56) was found in the Pamir-Alay [15,16]. Also, several chromosomal forms (2n = 52–48, NF = 56) were revealed in *E. alaicus* [17,18]. Moreover, the unique phenomenon of various types of meiotic contacts of non-homologous chromosomes (including fusions) forming dicentric chromosomes was discovered in *E. alaicus* [19].

For decades, *E. alaicus* had the status of a data-deficient species; its range was considered to be limited to a small area of the Alay Valley (Southern Kyrgyzstan). Recently, we demonstrated that this species is distributed not only throughout the Alay Valley but also in the valleys of the Naryn and Aksai Rivers (Kyrgyzstan), as well as on the territory of Tajikistan, in the upper Surkhob River area (Kyzylsu and Muksu River valleys) [18].

Here, we aimed to assess genetic variations by analysis of some mtDNA (*cytb,* cytochrome b gene) and nDNA genes (*XIST*, X-inactive specific transcript, and *IRBP*, interphotoreceptor retinoid-binding protein) as well as karyotype structure (differential G-banding) to study more thoroughly *E. alaicus* from the Alay Valley (the *terra typica* region, Kyrgyzstan), and Pamir-Alay (Tajikistan). We also evaluated the possible evolutionary input of natural hybridization between sibling species, *E. alaicus* and *E*. *tancrei*. We can suggest a more complex picture of genetic polymorphism in both species than is currently known, considering the extremely diverse landscape and climatic conditions of these mountainous regions.

## 2. Materials and Methods

### 2.1. Samples

Tissue samples and chromosome suspensions were obtained from mole voles collected during our field trips in 1981–1983, 2008, 2010, 2013, and 2015–2021. Information about the material used in the study is presented in Appendix A. The total number of specimens used for the molecular analyses was 60, from which 56 mole voles were karyotyped. Geographical locations of capture points of the mole voles are shown in Figure 1. The samples are deposited in the Joint Wild Animal Tissue Collection for Basic, Applied, and Conservation Research of the Core Centrum of the Koltzov Institute of Developmental Biology RAS, state registration number 6868145.

Animals were treated according to established international protocols, as in the Guidelines for Humane Endpoints for Animals Used in Biomedical Research. All the experimental protocols were approved by the Ethics Committee for Animal Research of the Koltzov Institute of Developmental Biology RAS in accordance with the Regulations for Laboratory Practice in the Russian Federation; the most recent protocol is numbered 37-25.06.2020. All efforts were made to minimize animal suffering.

### 2.2. Karyotyping

Chromosomes were prepared from bone marrow according to [20] for all animals listed with chromosome numbers in Appendix A. G-banding was achieved using trypsin digestion [21]. In total, we studied chromosomes for 59 specimens. Routine stained and G-banded metaphase plates were captured with a CMOS camera, mounted on an Axioskop 40 (Zeiss) microscope. Images were processed using Paint Shop Pro X2 (Corel).

### 2.3. Sequencing

Total DNA was isolated by phenol-chloroform deproteinization after treatment of shredded tissues (heart, muscle, or kidney) with proteinase K [22]. Primers used for amplification and sequencing of three overlapping fragments of the complete *cytb* gene [13,23], two overlapping fragments of the *IRBP* gene, exon 1 [24], and two non-overlapping fragments of the *XIST* gene [18] are listed in Appendix A. Polymerase chain reaction (PCR) was carried out in a mixture containing 25–35 ng DNA, 2 μL 10 × Taq-buffer, 1.6 μL 2.5 mM dNTP solution, 4 pM of each primer, 1 unit of Taq-polymerase (Syntol, Russia, Moscow), and deionized water to a final volume of 20 μL. Amplification was as follows: preheating at 94 °C for 3 min, then 35 cycles in a sequential mode of 30 s at 94 °C, 1 min at 55–67 °C depending on the applied pair of primers (see the Appendix A), and 1 min at 72 °C; the reaction was completed by a single final elongation of PCR products at 72 °C for 6 min. Automatic sequencing was carried out using a ABI PRISM^®^ BigDye^TM^ Terminator v. 3.1 Kit (Applied Biosystems, USA, MA, Waltham) with AB 3500 genetic analyzer at the Core Centrum of the Koltzov Institute of Developmental Biology RAS. All sequences have been deposited to the GenBank. Accession numbers were as follows: ON333901–ON333931 for the *cytb* gene, ON333848–ON333900 for the *IRBP* gene, ON314941–ON314990 for the *XIST* gene first fragment, and ON314885–ON314940 for the *XIST* gene second fragment. We also used some of the earlier published sequences [14,18] deposited by us to the GenBank (MG264318–MG264322, MG264324, MG264326–MG264330, MG264345, MG264346, MK544901, MK544903, MK544904, MK544906–MK544917, and MT468380 for the *cytb* gene; MT478768–MT478770 for the *IRBP* gene; and MK544921–MK544926 for the *XIST* gene first fragment; see the Appendix A). A total of 60 specimens were used for the mitochondrial *cytb* gene sequencing, with 56 specimens for the *IRBP* gene and the *XIST* gene sequencing.

### 2.4. Molecular Evolutionary Analyses

DNA sequences were aligned using the MUSCLE algorithm [25] in MEGA X software [26]. After alignment, the length of the *cytb* gene nucleotide sequence (determined in all mole voles) was equal to 1143 bp, the *IRBP* gene fragment was 1404 bp, and the two non-overlapping parts of the *XIST* gene comprised 444–448 and 525–526 bp (969–973 bp in total). Differences in the length of the latter gene fragments were related to deletions. The aligned fragments of the *XIST* gene were joined and analyzed as one sequence. The sequences of the protein-coding genes (*cytb* and *IRBP*) were presented by entire codons. Marginal codon boundaries (i.e., the first nucleotide of the initial codon and the third nucleotide of the last codon) in the *IRBP* gene fragment were determined from its sequences in genomes of house mice [27] and pine voles of the *Terricola* subgenus [28]. Sites in which two overlapping peaks were reproducibly registered in chromatograms were coded and treated as heterozygous.

Dendrograms were built based on the Maximum Likelihood (ML) method using IQTree software, version 2.0-rc2 [29,30]; the ModelFinder option [31] was applied to achieve optimal model evaluation of nucleotide substitutions for each gene. Before statistical analysis, non-extended deletions (revealed in the *XIST* gene sequences of several specimens) were filled in by random nucleotides which were absent in the total mole voles’ sample; this approach was used in analysis of the *XIST* gene variability in pine voles of the *Terricola* subgenus [28]. Final images of phylogenetic trees rooted on the genetically remotest group (i.e., mole voles from Mongolia) in all cases were rendered in FigTree 1.4.3 (http://tree.bio.ed.ac.uk/software/figtree/ (accessed on 25 November 2018)) and processed in Inkscape (https://inkscape.org/ (accessed on 27 September 2021)).

Along with phylogenetic analysis, haplotype networks (haplowebs) were built in the HaplowebMaker program (https://eeg-ebe.github.io/HaplowebMaker/ (accessed on 11 July 2020)) [32]. In the networks, the lengths of connecting lines correspond to genetic distances, whilst curves illustrate alleles that were found to be co-occurring in heterozygous individuals. In this way, a group of alleles linked together by heterozygotes represents an exclusive allele pool, and the corresponding groups of individuals are called a field for recombination (FFR). Such haplowebs have proven themselves to be useful in recent mammalian phylogenetic studies [33], but, as far as we know, they were tested on rodents for the first time in this work. The phasing of polymorphic loci in heterozygous genotypes was carried out based on combinations of alleles in the corresponding loci in homozygotes. For each case, we analyzed the known haplotypes of the surrounding populations to exclude random alternation of heterozygous sites [34].

## 3. Results

### 3.1. Chromosome Variability

New data include material collected from Gulcha vicinities in the north to the Pamir Highway in the south, as well as throughout almost the entire Alay Valley—from the areas bordering China in the east to Daroot-Korgon in the west. All the animals studied from this territory had 2n = 50–53. A re-collection was carried out near Gulcha (#1), where interspecific hybrids of *E. alaicus* and *E. tancrei* were discovered in 1983 [18]. Animals from there mainly had 2n = 52, with the exception of one female with 2n = 53, heterozygous for the *E. alaicus* species-specific chromosome Rb(2.11). This karyotype was identical to the one expected for the F1 hybrid of *E. tancrei* and *E. alaicus* (Figure 2a and [18]). All animals from the central and eastern part of the Alay Valley (#3–6) turned out to be the most typical homozygotes, with 2 Rb(2.11), 2n = 52. Further to the west, there is an interesting picture of chromosomal variability. Animals caught near Sary-Mogol (#7) in two cases had karyotypes 2n = 50, 2Rb(2.11), 2Rb(4.9), and in one case had 2n = 51, 2Rb(2.11), 1 Rb(4.9). A similar pattern was observed for samples from the vicinity of Daroot-Korgon, the westernmost collection points on the territory of Kyrgyzstan (#9, 10). Animals with 2n = 50, 2Rb(2.11), 2Rb(3.10) and 2n = 51, 2Rb(2.11), 1Rb(3.10) and a somatic mosaic with 2n = 50–51, 2Rb(2.11), 1-2Rb(3.10), were found there (Appendix A, Figure 2). Thus, to the south of Gulcha and in the Alay Valley, we have described different variants of the karyotype of the Alay mole vole with variations in Rb(2.11), Rb(3.10), and Rb(4.9).

Most of *E. tancrei* specimens studied in this work had 2n = 54; others exhibited several chromosomal forms with 2n = 30, 32, 34, with two animals from the contact zone of different chromosomal forms showing 2n = 51 and 52 (for descriptions see [16,35,36,37]). Unless these specimens of *E. tancrei* had the same 2n, as some *E. alaicus* (2n = 51–52), their karyotype structures were distinct due to non-homologous Rb translocations.

### 3.2. Sequencing Results

#### 3.2.1. *Cytb* Gene Variability

The complete *cytb* gene analysis, which was carried out by the Maximum Likelihood (ML) approach, revealed no the specific clusters for two *Ellobius* species (Figure 3). Specimens of *E. tancrei* from Mongolia form the more distant clade in relation to all other mole voles of both species, whose haplotypes are distributed in two closer sister clades in accordance with species belonging. The similar results were described earlier, and mole voles from western and central parts of Mongolia were proposed to belong to a new cryptic species; however, these individuals are currently considered as *E. tancrei* s. str. [14]. Within the main *E. tancrei* clade, joining mole voles from the Central Asia, specimens from Uzbekistan (#27) and from the southwestern regions of Tajikistan, remote from the Surkhob and Alay Valleys (#26), appeared to be isolated. In the *E. alaicus* clade, the animals from Gulcha vicinities (#1), which turned out to be basal, attract attention. Along with them, two mole voles from the Taldyk Pass (#3), both individuals from Taunmuruk (#5), and one mole vole from Sary-Tash vicinities (#6)—which have three common substitutions, unique for the total mole vole sample—differ from other examples to the greatest extent. Another single substitution distinguished all Alay mole voles from Kyrgyzstan (the eastern group on ML-tree) from representatives of the same species from Tajikistan (the western group). However, in general, *cytb* gene variability in *E. alaicus* and in Central Asian populations of *E. tancrei* is weakly expressed and demonstrates no clear relation with chromosomal polymorphism.

On the haplotype networks, shades of green and blue mark haplotypes typical of *E. tancrei*, and shades of purple and red-yellow show haplotypes typical of *E. alaicus* (see Figure 4a). A haploweb was built on the basis of data on *cytb* and visualizes the genetic distances between clades and the pattern of nucleotide polymorphism. In particular, it demonstrates that the Mongolian population of *E. tancrei* is separated from the rest of the studied mole voles by a greater genetic distance than that between Central Asian *E. tancrei* and *E. alaicus*. The complex, mesh structure of *E. alaicus* haplotypes and its lower resolution compared to Central Asian *E. tancrei* should also be noted.

#### 3.2.2. *XIST* Gene Variability

Analysis of the *XIST* gene variability demonstrated differentiation *E. tancrei* to some intraspecific groups: from Mongolia, Tashkent, South-Western Tajikistan, and the rest of the populations from Tajikistan (Figure 5). As in the case with the *cytb* gene, mole voles from Mongolia formed the most remote clade in the ML-tree, and other clades and branches (including ones related to the Alay mole vole) composed a star-like structure. Unexpectedly, we revealed two distinct lineages in the Alay mole vole, using the *XIST* gene. The first variant of this gene was more widely distributed; it was registered in specimens from almost all studied parts of *E. alaicus* range, and in Tajikistan populations, it was the single. Another *XIST* variant that was firstly discovered by us in the work significantly differs from the first variant by several substitutions and three-nucleotide deletion near the start of the analyzed sequence. This was diagnosed in many Kyrgyzstan populations of mole voles, from Gulcha vicinities to Daroot-Korgon, both in homozygous (specimens 27353, 27487, 27488, 27493, 27494, 27500, and 27505) and heterozygous states (specimens 27495, 27496, 27351, 27497, 27489, 27491, 27354, and 27498). Mosaic and wide spreading of the variant across the Alay Valley indicates that it is specific for *E. alaicus* rather than for *E. tancrei.* Genetic *p*-distance between two variants of the *XIST* gene of *E. alaicus* (0.003) is similar with *p*-distances between the common, more distributed variant of *E. alaicus* and variants of Central Asian *E. tancrei* specimens (0.003–0.004), which makes species determination by the *XIST* gene fragments more complex.

Conventionally, the common *XIST* gene variant can be designated as “Western” (W) and variant with deletion as “Eastern” (E), since homozygous individuals with the W haplotype were not found to the east of Sary-Mogol (#7), and the E-variant was not found (either in homozygous or heterozygous states) to the west of Daroot-Korgon.

Identity of homozygous genotypes (including ones with a deletion) in Alay mole voles from different localities allowed us to propose its invariability in all mole voles from the Alay Valley and adjacent territories and determine a hypothetical second haplotype in heterozygous specimens, “deducting” the haplotype with deletion from heterozygous sequences. In almost all mole voles, heterozygous genotypes were presented by a haplotype common in *E. alaicus*, and a haplotype with deletion. However, in three mole voles from Gulcha vicinities (27491, 27497, and 27498), the heterozygous genotypes included a haplotype with deletion and a tancrei-like haplotype (Figure 5). This result supports the idea of a hybrid zone between *E. alaicus* and *E. tancrei* in this locality, but further study using other nuclear genes is required.

On the haplotype network (‘haploweb’), two haplotypes were clearly distinguished within *E. alaicus* too (Figure 4b). Five hybrid animals—from the Taldyk pass (#3), Sary-Tash and its vicinities (#4, 6), and Taunmuruk (#5)—were shown in arcs. At the intersection of the network lines, a blue dot indicated a possible hybrid haplotype of mole voles from the Gulcha vicinities (#1), the hybrid zone described earlier [18]. As can be seen from the length of the lines corresponding to the genetic distances, this haplotype is shifted closer to *E. tancrei*.

#### 3.2.3. *IRBP* Gene Variability

The ML-dendrogram and haplotype network built for the *IRBP* gene significantly differed from reconstructions based on other markers. Three main clades, which formed a star-like structure, are obvious in the ML-dendrogram. The first clade included mole voles from Mongolia, the second clade included all Central Asian *E. tancrei* and some *E. alaicus* specimens, and the third one included the majority of Alay mole voles (Figure 6). Genotypes which compose the clades presented by Mongolian and Central Asian *E. tancrei* and the clade joining almost all *E. alaicus* specimens were characterized by three fixed substitutions; for several more sites, the differences between these groups were traced, but they were not absolute because of mole voles from Mongolia and Tashkent. One of the clades (including almost all Alay mole voles, both from Tajikistan and the Alay Valley) was actually presented by the same variant of the *IRBP* gene (excluding one specimen, which was heterozygotic for one site). The clade including Central Asian *E. tancrei* individuals was significantly more variable; up to six distinct subclades and branches may be traced in it. Among these specimens, the mole vole from Tashkent differed to the greatest extent. It is extremely interesting that three Alay mole voles (two specimens from Sary-Mogol (27493 and 27503) and one from Taunmuruk (27495)) and two mole voles from proposed hybrid zone of *E. alaicus* and *E. tancrei* in Gulcha vicinities (27491 and 27497) came to join this clade, demonstrating no differences from *E. tancrei* specimens from Aivadj, Sarinai, and Obi-Kaboud (Tajikistan). Moreover, along with homozygous genotypes, several mole voles (simultaneously heterozygous for all three sites) demonstrated fixed differences between two main clades (two specimens from Gulcha vicinities (27488 and 27498) and two mole voles from Daroot-Korgon, point 2). The presence of two genetically discrete groups, the invariability of one of them, and the lack of intermediate variants allowed us to divide heterozygous genotypes into haplotypes; a similar approach was successfully used earlier for determination of haplotype combination in hybrid house mice [38]. Haplotypes which were hypothetically determined in heterozygous mole voles occupied the same positions in the ML-tree as different homozygous genotypes.

The haplotype network built according to the *IRBP* gene sequences (Figure 4c) demonstrated low polymorphism; sibling species were represented by two haplotypes, which corresponded to almost all animals from the entire sample. The exceptions were some populations of *E. tancrei*: Shilbili (#21), Tashkent (#27), and Mongolia (#28–30). The populations from Tashkent and Mongolia occupied an intermediate position on the net between *E. tancrei* and *E. alaicus*. It is important to note that the ‘pure’ *E. tancrei* haplotype was presented in several individuals identified by other characters as *E. alaicus*, the same as in the ML-tree. These were animals from Taunmuruk (#5) and Sary-Mogol (#7). Hybrid genotypes were found in specimens from Daroot-Korgon (#9) and Gulcha (#1).

#### 3.2.4. Multigenic Haplowebs

A consensus network based on nuclear genes (*XIST + IRBP*) demonstrated the compactness of a group of *E. tancrei* populations from Tajikistan (Figure 7a). Only the southern locality of Aivaj (#26) and the eastern one of Shilbili (#21) showed slightly different haplotypes. *E. tancrei* individuals from Tashkent and Mongolia turned out to be closer in genetic distances to interspecific hybrids from the Gulcha vicinities (#1). Combining data on two nuclear genes allowed us to evaluate a complex system of *E. alaicus* hybrids—interspecific with *E. tancrei* as well as intraspecific between “E” and “W” lineages. Special attention should be paid to the mole voles from Gulcha vicinities (#1)—the scale of polymorphism in this locality and the high degree of differences between the involved haplotypes becomes visible on this network. As can be seen, in this locality, there is a single di-heterozygote for the two nuclear genes studied (specimen 27488).

The consensus network for three genes, including data on *cytb*, had a higher resolution due to mitochondrial polymorphism (Figure 7b). Three main branches are clearly distinguished on it—the Central Asian *E. tancrei* (from Tajikistan and Uzbekistan), the Mongolian *E. tancrei,* and *E. alaicus*. There was a high degree of genetic polymorphism within *E. alaicus* and an abundance of hybrids of varying degrees of remoteness. There was also a complex pattern of hybridization in individuals from Taunmuruk (the easternmost point of the collections (#5)) despite the extremely small representation of this locality (only two individuals). It should also be noted that there were no hybrid forms among individuals from Muksu (#15) to Achek-Alma (#11), where the chromosomal form 2n = 48 was described.

## 4. Discussion

The genetic variability of *E. alaicus* in the Alay Valley (the large territory originating at the junction of Tien Shan and Pamir Mountain systems) appeared to be unexpectedly high and complicated due to mito-nuclear discordances. Our results, which were obtained from the analyses of karyotypes and molecular markers, revealed that distinctive variability ‘layers’ and the spectra of polymorphism partly coincide with each other.

*E. alaicus* demonstrated variable chromosome numbers (2n = 48–52) with four variants of Robertsonian translocations: Rb(2,11), Rb(1,3), Rb(4,9), and Rb(3,10) [18]. The chromosome Rb(2.11) was found in all *E. alaicus* (with one exception for interspecific hybrids outside the Alay Valley, Naryn region) and some forms of *E. tancrei*, in which, as we hypothesized [16,18], this translocation originated independently from *E. alaicus*. Therefore, the translocation is considered typical for *E. alaicus*. Here, using routine staining and G-banding, we revealed six cytotypes in the studied specimens. We confirmed 2n = 52, NF = 56, XX—typical for *E. alaicus* in the new points (see Appendix A, #3–6, and Figure 1). We also found a heterozygous specimen (2n = 53, NF = 56; 27491; see Appendix A, #1, and Figure 1 and Figure 2a) at a point close to the area previously described as a contact zone with *E. tancrei* [18]. The most fascinating issues were new chromosomal variants (2n = 50–51) in points located in the western part of the Alay Valley. It should be emphasized that the decrease in the diploid number was due to the appearance of two Robertsonian translocations (Rb(3.10) and Rb(4.9)) which had previously been observed in samples from the Pamir-Alay (see Appendix A, #11) [18]. These Rbs were revealed in homo- and heterozygous variants (see Appendix A, #9, 10, and Figure 2). Thus, all verified translocations were previously described for *E. alaicus* [18].

Concerning molecular data, the *cytb*, *XIST*, and *IRBP* genes provided phylogenetic reconstructions with different topologies, various genetic distances, and interpositions of clades related to intraspecific forms of *E. tancrei* and *E. alaicus.* It is very interesting that, in contrast to the general belief that the mitochondrial *cytb* gene is evolving more rapidly, the nuclear *XIST* gene allowed us to distinguish two cryptic lineages within *E. alaicus*, and the use of the *IRBP* gene revealed the typical *E. tancrei* haplotype to be present in some Alay mole voles from those localities, where it would be difficult to propose interspecific hybridization. Thus, the most noticeable and debatable result of the molecular data analysis appeared to be their inconsistency with chromosomal data, as well as discrepancies in phylogenetic reconstructions for different markers. The generalization of the results obtained here is a non-trivial task, since it requires accurate information about the modern and historical ranges of mole voles, as well as a deep analysis of their genomes. Unfortunately, there are many blank spots in these areas, so some conclusions remain at the level of hypotheses awaiting further elaboration. Nevertheless, we consider it possible to note the following facts.

Phylogenetic reconstruction based on the *cytb* gene confirmed the proposed distribution of individuals between species. Morphologically and cytogenetically detected *E. alaicus* and Central Asian *E. tancrei* corresponded to two distinct clades in the *cytb* ML phylogenetic tree (Figure 3). The consistent distinctiveness of *E. tancrei* samples from Mongolia may indicate the need to clarify the phylogenetic and taxonomic status of the Mongolian populations of the eastern mole voles. The structure of the Central Asian *E. tancrei* clade, being weakly expressed, nevertheless reflected the geographic distribution of the samples. All animals originating from the same locality were clustered together. In contrast, in the clade *E. alaicus*, animals from the same localities were often distributed into different subclusters. It is noteworthy that these were individuals from parts of the range remote from the contact zone with *E. tancrei,* namely, the localities Daroot-Korgon (# 8–10) and the pass Taldyk (#2, 3). Such variety suggested that these samples were carriers of ancestral polymorphism and/or secondary intensive hybridization of *E. alaicus* intraspecific forms. In the case of mole voles from Gulcha vicinities (#1), the interpretation of their basal position in relation to the other haplotypes in *E. alaicus* clade is ambiguous. On the one hand, although this locality is situated in the previously described zone of contact and hybridization with *E. tancrei*, and we found one specimen with the hybrid karyotype (27491, Figure 2a), haplotypes of mole voles from Gulcha clearly belonged to the *E. alaicus* clade. On the other hand, overall, the *E. alaicus* haplotypes were most closely related to Central Asian *E. tancrei*, which cannot be assertively interpreted as haplotype change due to interspecific hybridization. Possibly, territories to the north from the Alay Ridge were the place of *E. alaicus* origin. Although the single substitution supported to some extent the separation of *E. alaicus* populations from Tajikistan and the Alay Valley (i.e., 48 chromosomal form from others), specimens with 2n = 50–52 did not show clear correspondence to the *cytb* gene data.

We assumed that for the reconstruction of the sibling species phylogeny, data on the *XIST* gene (one of the actively used sex-linked markers) would be valuable [14,18,28,39]. Indeed, the analysis of the sequences of its two fragments transformed and added details to the above picture. Even though the phylogenetic signal is rather weak, the *XIST* data confirm a number of general trends. As can be seen from the haplotype network (Figure 4b), *E. tancrei* and *E. alaicus* again turned out to be isolated, with interspecific hybrids near Gulcha (#1). It is easy to notice that most *E. tancrei* individuals had one typical haplotype, from which only the sequences of animals from Mongolia (#28–30), Uzbekistan (#27) and southern Tajikistan (#26) differed. It is noteworthy that samples of *E. tancrei* from Mongolia and Uzbekistan turned out to be closer to interspecific hybrids from Gulcha vicinities than to the majority of Central Asian *E. tancrei* specimens.

Within *E. tancrei*, the main clades (obtained with the use of the *XIST* gene fragments) in general corresponded to ones in reconstructions built on the *cytb* gene; however, within *E. alaicus*, two cryptic lineages were clearly distinguished. Genotypes and haplotypes with deletion revealed in many localities of the Alay Valley seem to be evolutionarily new because analogous variants have not been registered in any species of the subgenus *Ellobius* yet. Taking into consideration significant differences between the common genotype of *E. alaicus* and the genotype with deletion as well as an absence of intermediate variants, we presumed their independent origin and evolution in different areas of the Alay Valley and the Pamir-Alay. Factors which lead to the separation of the two *E. alaicus* lineages remain unknown; intensive development of glaciers might be such unfavorable circumstances. Later, when fit conditions had been restored, and Alay mole voles began to actively occupy new areas, the two lineages might have entered introgressive hybridization and formed mixed populations on almost all territories of the Alay Valley. The ‘mosaic’ character of the *XIST* gene variability in *E. alaicus* was similar to the *cytb* gene polymorphism. If we are to assume that factors which lead to similar variability of both the genes are the same, the idea of two discrete intraspecific forms hybridized in *E. alaicus* seems to be preferable over the idea of wide ancestral polymorphism presence. The absence of the *XIST* haplotype with a deletion in *E. alaicus* populations from Tajikistan confirmed their isolation from populations from the Alay Valley, similarly to the results of *cytb* gene analysis.

ML-reconstruction based on the *IRBP* gene analysis showed the most unexpected structure of phylogenetic relationships. Some *E. alaicus* individuals had ‘pure’ *E. tancrei* genotypes for this gene and were placed in the corresponding clade. These were animals from different parts of the Alay Valley remote from each other (Sary-Mogol, #7; Taunmuruk, #5) and from the hybrid zone (Figure 6). ‘Hybrid’ genotypes were found in individuals from Gulcha vicinities (#1), which is not surprising; however, the same genotypes were revealed in two specimens from Daroot-Korgon (#9), the locality in the Alay Valley most distant from Gulcha vicinities (#1). No presumable hybrids were found with the use of the *IRBP* gene at the border of the ranges of the two species in north-eastern Tajikistan, in the Muksu-Shilbili-Kashat-Utol-Poyon region: there was neither *E. alaicus* with the *E. tancrei IRBP* genotype, nor vice versa. At the same time, such animals were found in remote parts of the range of the Alay mole vole, where we did not previously assume recent contact between *E. alaicus* and *E. tancrei*. The haplotype network demonstrated only two haplotypes for the sibling species, which corresponded to almost all animals from the sample, and highlighted hybrid interspecific genotypes in specimens from Daroot-Korgon (#9) and Gulcha vicinities (#1).

There are two possible explanations for this paradox.

The first possible interpretation is based on the existence of ancestral polymorphism. A clear division of mole voles by the *IRBP* in the contact zone in Tajikistan and the preservation of the ‘pure’ *E. tancrei* haplotype to the east, within the range of *E. alaicus*, fit the hypothesis of ancestral polymorphism. As we noted above, *E. alaicus* populations, habiting territories to the north of the Alay Ridge, may be considered as ancestral in comparison with *E. alaicus* from the Alay Valley and North-Eastern Tajikistan. Thus, it is possible that some Alay mole voles, which settled further to the east, carry the ancestral *IRBP* gene variants in their genotypes. This proposition is supported by chromosomal data, too. Western populations of *E. alaicus*, from Muksu (#15) to Achek-Alma (#11), had 2n = 48 and three Rb translocations in the homozygous state [18]. The absence of interspecific hybrids noted according to molecular data in the Muksu-Shilbili-Kashat-Utol-Poyon area indicated a reliable reproductive barrier between *E. alaicus* with 2n = 48 and *E. tancrei* with 2n = 54. The populations of *E. alaicus* from the territory of Kyrgyzstan possessed 2n = 50–53 were karyotypically closer to *E. tancrei* than the western ones, which was also confirmed by all molecular markers to varying degrees. Nevertheless, there are contradictions in the details. For example, in Daroot-Korgon (#9) specimens had different karyotypes (27490, 2n = 50, 2Rb(2.11), 2Rb(3.10) and 27499, 2n = 51, 2Rb(2.11) 1Rb(3.10)), but were identical in the analyzed gene sequences. In the part of the range of *E. alaicus*, considering there to be an active process of speciation, we therefore inevitably encounter incomplete lineage sorting and ancestral polymorphism.

However, there are some drawbacks to this explanation. While *E. alaicus* has its own *IRBP* genotype, distributed across almost all the species range, it is not likely that some *E. alaicus* specimens in several populations would keep an ancestral genotype without any changes. Moreover, ancestral polymorphism is expected to be more widely distributed, but the extrinsic *IRBP* variant was found only in two localities in the western part of the Alay Valley and at the easternmost point of the collections, in Taunmuruk. The presence of characteristic *E. tancrei IRBP* genotypes and haplotypes in genomes of some *E. alaicus* individuals can also be explained by recent secondary contacts and hybridization. However, this assumption explains well only the presence of interspecific hybrids. It remains unclear where *E. tancrei* specimens come from. Still, we do not exclude the possibility of *E. tancrei* migration across the Alay or Ferghana Ridges along mountain canyons and even passes (we found colonies of mole voles at an altitude of about 4000 m above sea level; see Figure 1). In particular, we have previously shown that the Ferghana Ridge delimits the ranges of two species of mole voles [18]. It is possible that *E. tancrei* living on the northern slopes settled along the small river valleys (Isfairam-Sai, Kek-suu, etc.); overcame the ridges; and went south to Daroot-Korgon, Sary-Mogol, and Taunmuruk (Figure 1, #5, 9, 10).

Experimental hybridization in sibling species of the subgenus *Ellobius* was studied earlier for *E. tancrei* and *E. talpinus*. We only obtained the F1 generation, and revealed the cytogenetic mechanisms which corrupted meiotic nuclear architecture. Formations of chromosome chains, stretched centromeres, and distorted recombination in meiotic prophase I lead to hybrid sterility [40]. In another study, we obtained hybrids of *E. tancrei* and *E. alaicus.* In this case, chromosome chains due to partial homology of Rbs were also found in meiosis of hybrids, but their presence only led to low fertility of the first generation; in the later generations, the fertility gradually restored [41]. We supposed that in contrast to *E. tancrei* and *E. alaicus,* the second pair of species (*E. tancrei* and *E. talpinus*) possessed complete reproductive barriers. Apparently, in the contact zone, each species maintains the integrity of the gene pool, which is an essential species criterion according to the concept of biological species [42]. There were no data for *E. tancrei–E. talpinus* contact zones in nature. In the case of *E. tancrei–E. alaicus*, we described a contact zone previously [18]. One more assumption might be that within the range of *E. alaicus* exists currently undiscovered refugia of *E. tancrei*. The expansion of possible contact zones could significantly affect the gene pool of the surrounding *E. alaicus*, for example, in the Sary-Mogol region and Taunmuruk (Figure 1, #5, 7). Even if we have not observed any mixed colonies of the two species in the studied area in southwestern Kyrgyzstan, the hypothesis that hybridization took place in the rather recent past can be suggested.

Implementation of karyological data along with molecular markers has added an important layer to the multidimensional picture of the evolution of mole voles. Comparing molecular and chromosomal data, we again turn to the hypothesis of ancestral polymorphism. Western populations of *E. alaicus*, from Muksu to Achek-Alma (Figure 1, #11–15), have 2n = 48 and carry three Rb translocations in the homozygous state. We were unable to find interspecific hybrids in the Shilbili-Kashat-Muksu-Utol-Poyon area between *E. alaicus* with 2n = 48 and *E. tancrei* with 2n = 54. The eastern populations of *E. alaicus* from Alay Valley have 2n = 50–53 and demonstrate unevenness of Rbs distribution, with a high number of specimens with heterozygous chromosomal sets. The meiotic chromosome contacts detected in *E. alaicus* [19] can be considered as plausible mechanisms for Rbs origin and might explain both the formation of Rb(2.11) (which is typical for this species) and the rapid de novo formation of the rearrangements that we detected throughout the species range [18]. We are probably dealing with an active process of speciation taking place before our eyes, and therefore we inevitably encounter incomplete lineage sorting, ancestral polymorphism, and chromosomal speciation.

Thus, in both mole vole species, karyotypic changes did not fully correspond to nDNA marker (fragments of *XIST* and *IRBP* genes) variability, which, in turn, differed in their patterns both each from other and from the *cytb* variability. This genetic discordance in the subterranean mole vole *E. alaicus* confirmed our earlier suggestion that this species undergoes rapid evolution. An analogous situation was observed in a sibling species of mole voles, *E. tancrei*. The character of *E. alaicus* and *E. tancrei* intraspecific differentiation, which is significant and territorially wide, indicates that it might originate due to habitat fragmentation. High altitudes (2500 m and over) and a continental climate, characterizing this cold semi-arid region distinguished by low precipitation (average ca. 200–400 mm per year) [43], contribute to the sparse vegetation. Mole voles mostly feed on roots of ephemeroid plants, so the distribution of these animals is mosaic and highly correlated with vegetation. Separation of the range into small isolated populations creates conditions for accelerated evolutionary processes because of low effective population sizes and isolation. In result, in isolated populations, different mutations might occasionally fix in chromosomes and mitochondrial or nuclear genes; that is, ‘evolutionary potential’ may be realized by diverse ways. This factor together with possible hybridization of intraspecific forms and species has resulted in the complicated genetic variability of *E. alaicus* and *E. tancrei*.

Fixation of new chromosomal re-arrangements seems to be common in species of the genus *Ctenomys* (2n = 10–70) and, as a rule, is not accompanied by sterility, decreased fitness, or negative heterosis of heterozygous carriers [44]. The hypothesis of the leading role of chromosomal evolution in *Ctenomys* [45] was not supported by mtDNA and microsatellite data. Attempts to find a correlation between morphological and chromosomal variability in *Ctenomys* have also not been successful [46,47].

Similar examples of ‘mosaic’ intraspecific polymorphism were described in African mole rats of the genus *Fukomys*, which are hypervariable in karyotypic characteristics and demonstrate relatively small polymorphism in molecular markers, and vice versa [48]. This example is important, since mole rats of the genus *Fukomys* have amazing convergent similarity with mole voles of the genus *Ellobius* in a number of characteristics: subterranean living, morphological features, the abundance of chromosomal forms, and the complex social structure of families.

One of the most striking examples is a group of pine voles of the subgenus *Terricola* (genus *Microtus*), which includes several species with different patterns of chromosomal polymorphism. As demonstrated in a recent work involving molecular markers [28], in the Caucasus pine vole *M. daghestanicus*, along with a wide Robertsonian chromosome fan, significant variability of mitochondrial *cytb* gene and even nuclear *BRCA1* gene was observed, where their different variants existed in the same chromosome form and even the same population. It should be noted that the wide molecular polymorphism was explained by ecological peculiarities of Caucasus pine voles, inhabiting alpine meadows (likely mole voles) and evolving in small isolated populations. Along with the Caucasus pine vole, significant DNA variability was discovered in common pine voles (*M. subterraneus*) from Asia Minor, although they belong to the same chromosomal form.

It is likely that such examples of inconsistent evolutionary rates of different traits, in particular, morphology, chromosomes, and the structure of nDNA and mDNA genes, could be indicative for the explosive nature of speciation in these groups. Subterranean living, limited mobility, and mosaic habitats could drive rapid fixation of genetic changes, which we may consider as a discrepancy simply because we describe a snapshot of the speciation process.

## 5. Conclusions

Our results, despite the seeming inconsistency, perfectly illustrate the processes accompanying explosive genetic diversification in such a unique model of subterranean rodents represented by mole voles. At each of the studied genome levels, we observed a continuum of different forms, chromosomal and molecular, whose origin was apparently due to extremely varied conditions and habitat fragmentation in mountains. It should not be excluded that the exclusive feature of *E. alaicus* chromosomes, ‘contacts first in meiosis’, was one of causes of karyotype variability in this species.

Phylogenetic reconstructions based on some nuclear and mitochondrial loci allowed us to observe diverse ways in which the species gene pool can change: a clear division into *E. alaicus* and *E. tancrei* species according to *cytb* and signs of ancestral polymorphism or hybridization according to nuclear markers. Possible hybridization in the secondary contact zone could lead to the presence of heterozygotes and even homozygous genotypes, characteristic for the sibling species *E. tancrei,* in the distinct parts of the *E. alaicus* range. These processes created the most complex geographical mosaic on the studied part of the area. However, they are a kind of “time sweep” of the evolutionary process, which we can rarely observe on species with lower rates of evolution in less specific habitat conditions.

## Figures and Tables

**Figure 1 life-12-00728-f001:**
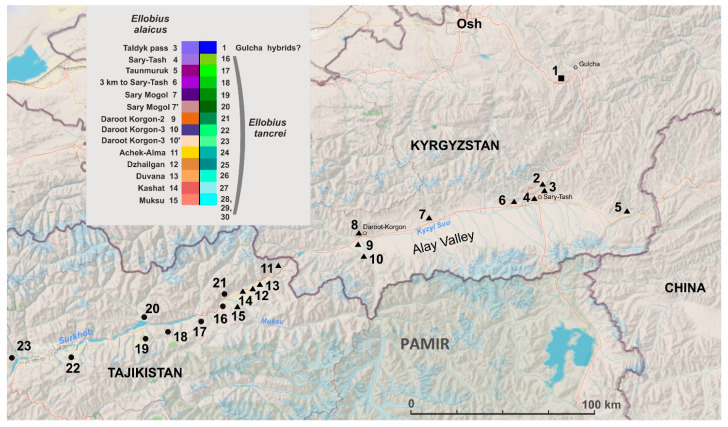
Map of localities studied for *E. alaicus* (triangles), *E. tancrei* (circles), and proposed hybrid zone (square) in Gulcha vicinities (Kyrgyzstan). Colors of the incorporated table correspond to the numbers of studied localities (see Appendix A). Points 24–30 were beyond the map.

**Figure 2 life-12-00728-f002:**
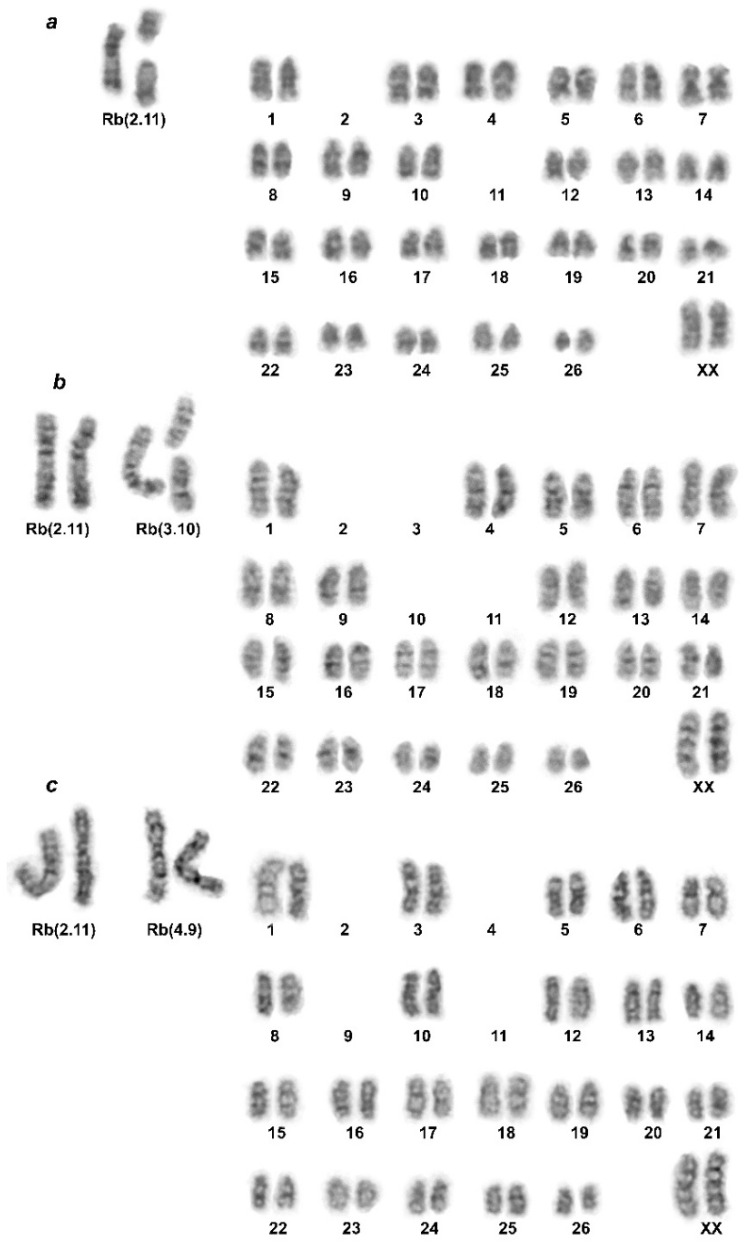
Chromosomes of *E. alaicus* and a hybrid with *E. tancrei*, G-banded karyotypes: (**a**) 27491, 2n = 53, hybrid, #1; (**b**) 27487, 2n = 51, #10; (**c**) 27503, 2n = 50, #7.

**Figure 3 life-12-00728-f003:**
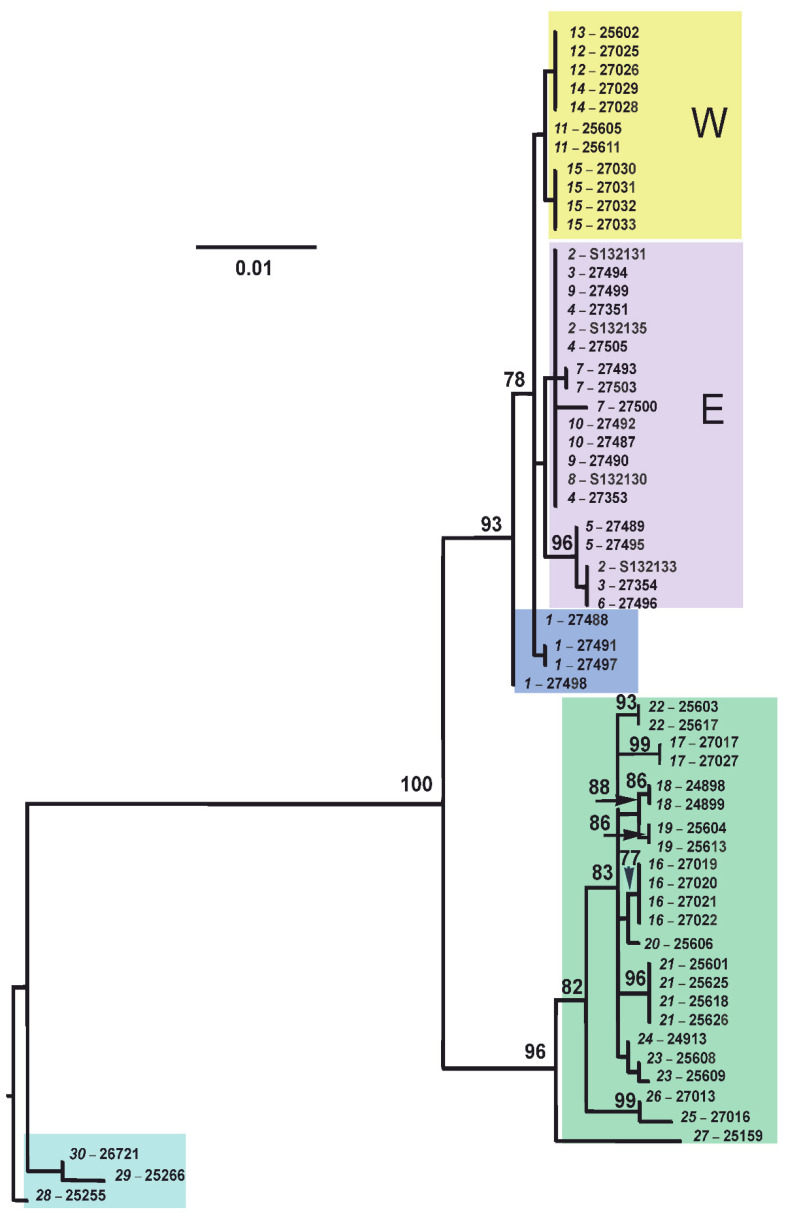
The maximum-likelihood tree as inferred from the *cytb* gene data on 60 specimens of *E. tancrei* and *E. alaicus*. The dendrogram was rooted on the group of mole voles from Mongolia. Numbers above nodes correspond to bootstrap support (>70%). The names of the samples are presented in the form of “locality number—individual number”. The color selection corresponds to haplowebs: cyan marks Mongolian *E. tancrei* (#28–30); green marks Central Asian *E. tancrei*; blue marks probable interspecific hybrids from Gulcha vicinities (#1); purple and yellow mark “eastern” and “western” haplotypes of *E. alaicus*, respectively.

**Figure 4 life-12-00728-f004:**
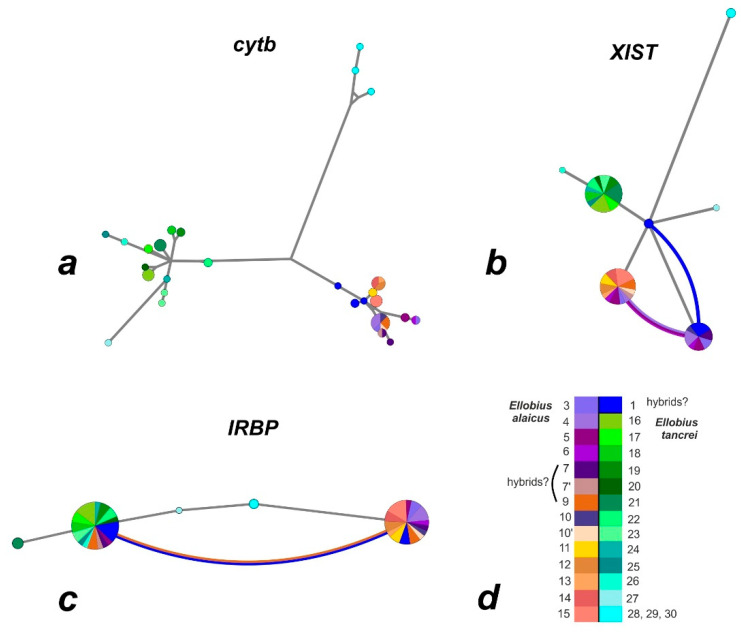
Haplowebs generated by HaplowebMaker for *cytb* (**a**), *XIST* (**b**), and *IRBP* (**c**) genes of studied mole voles (see Appendix A). The circles represent alleles with sizes proportional to their frequency in the populations. The investigated taxonomic hypotheses are distinguished by different colors (**d**). The length of the branches is proportional to the number of mutations. The curves connect individuals sharing one unique pool of alleles. The thickness of the curves is proportional to the number of hybrids.

**Figure 5 life-12-00728-f005:**
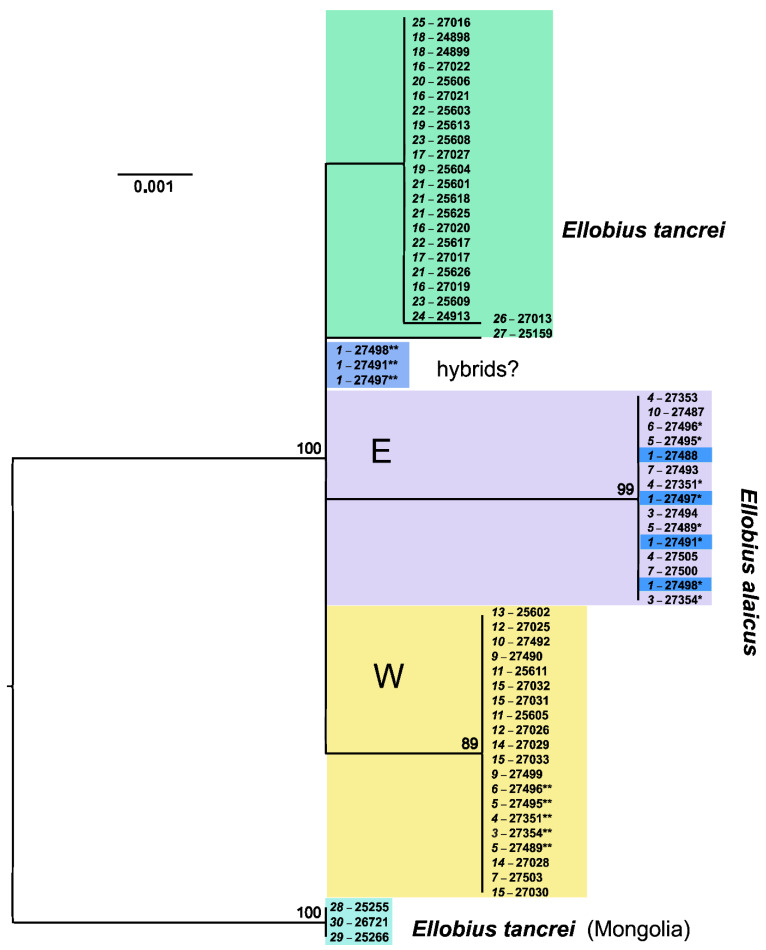
The maximum-likelihood tree as inferred from the *XIST* gene data on 56 specimens of *E. tancrei* and *E. alaicus*. The dendrogram was rooted on the group of mole voles from Mongolia. Numbers above nodes correspond to bootstrap support (>70%). The names of the samples are presented in the form of “locality number—individual number”. Haplotypes, which were hypothetically deduced from heterozygous genotypes, are designated by asterisks: the new haplotype with deletion is marked by one asterisk, whilst another haplotype is marked by two asterisks. The color selection corresponds to haplowebs: cyan marks Mongolian *E. tancrei* (#28–30); green marks Central Asian *E. tancrei* specimens; blue marks probable interspecific hybrids from Gulcha vicinities (#1); purple and yellow mark “eastern” and “western” haplotypes of *E. alaicus*, respectively.

**Figure 6 life-12-00728-f006:**
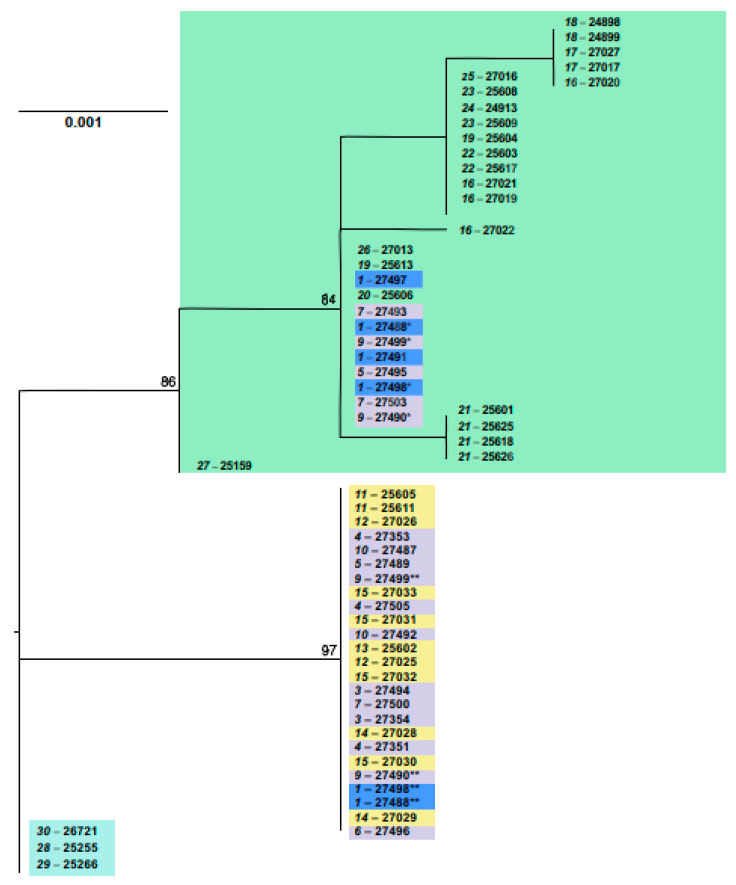
The maximum-likelihood tree as inferred from the *IRBP* gene data on 56 specimens of *E. tancrei* and *E. alaicus*. The dendrogram was rooted on the group of mole voles from Mongolia. Numbers above nodes correspond to bootstrap support (>70%). The names of the samples are presented in the form of “locality number—individual number”. Haplotypes, which were hypothetically deduced from heterozygous genotypes, are marked by asterisks: the haplotype typical for *E. tancrei* is marked by one asterisk, whilst the haplotype typical for the majority of *E. alaicus* specimens is marked by two asterisks. The color selection corresponds to haplowebs: cyan marks Mongolian *E. tancrei* (#28–30); green marks Central Asian *E. tancrei* specimens; blue marks probable interspecific hybrids from Gulcha vicinities (#1); purple and yellow mark *E. alaicus* specimens with “eastern” and “western” *XIST* haplotypes, respectively.

**Figure 7 life-12-00728-f007:**
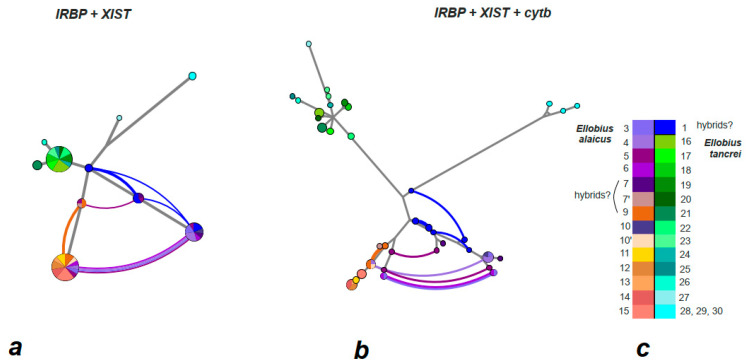
Haplowebs for two nuclear markers (**a**) and three markers, including *cytb* (**b**), of studied mole voles (see Appendix A). The investigated taxonomic hypotheses are distinguished by different colors (**c**). The circles represent alleles with sizes proportional to their frequency in the populations. The length of the branches is proportional to the number of mutations. The curves connect individuals sharing one unique pool of alleles. The thickness of the curves is proportional to the number of hybrids.

## Data Availability

GenBank accession numbers were as follows: ON333901–ON333931 for *cytb* gene, ON333848–ON333900 for *IRBP* gene, ON314941–ON314990 for *XIST* gene first fragment, and ON314885–ON314940 for *XIST* gene second fragment.

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
