# Peer review of "Geographic Mosaic of Extensive Genetic Variations in Subterranean Mole Voles Ellobius alaicus as a Consequence of Habitat Fragmentation and Hybridization"

_life, 2022, doi:10.3390/life12050728_

Round 1

Reviewer 1 Report

On the whole, the presented manuscript is a very interesting work, useful for both cytogeneticists and zoologists, and providing new data and insights into chromosomal and molecular evolution in Ellobius mammals.

There are two main concerns regarding this MS.  First, in the section Materials and Methods, as well as in the captions to the figures, nothing is said about the rooting of trees. If phylogenetic trees were rooted by using the Mongolian lineage of E. tancrei as an outgroup, then it (the Mongolian lineage) cannot be considered as a basal line, since it represents the fixed outgroup. If the trees were not rooted (and it seems that they were not rooted), then the direction of evolution is unknown, and by definition, no branch can be considered as a “basal” one. An obvious solution to this problem is the inclusion in the phylogenetic  analysis of another, obviously external group, which can be used as a formal out group.

Secondly, there is a too big gap between the Results sections, in which all the findings are presented very punctually, and the Discussion, in which the conclusions are too general. Because of this, in some places it is difficult to follow the logic of the authors. Let me explain this. So, it is said that there is a mito-nuclear discordance. The idea is clear, but, it would be useful to summarize at the beginning of the Discussion section how this discrepancy manifested itself (which individuals/populations produced different topologies when using different genes).

It should also be clearer explained on which basis the population (individuals) from Gulcha (locality 1) are positioned as hybrids.

In the Abstract the sentence “Analysis of IRBP demonstrated the presence of the specific genotype in most of E. alaicus specimens, but also revealed typical for E. tancrei haplotype” is not clear. Maybe the authors meant “Analysis of IRBP revealed not only haplotypes specific for E. alaicus, but also haplotypes shared by E. alaicus and by E. tancrei”

The scheme (Figure 1) lacks some important points, for example the locality 2 (Taldyk pass).

 I strongly recommend accepting the work for publication after the proposed revision has been made.

Author Response

We greatly appreciate the time and efforts you’ve put into your comments. Your advice on the methodology and pictures of our paper are very helpful. The comments greatly improved the consistency of the text and made it easier to understand.

  1. First, in the section Materials and Methods, as well as in the captions to the figures, nothing is said about the rooting of trees. If phylogenetic trees were rooted by using the Mongolian lineage of E. tancrei as an outgroup, then it (the Mongolian lineage) cannot be considered as a basal line, since it represents the fixed outgroup. If the trees were not rooted (and it seems that they were not rooted), then the direction of evolution is unknown, and by definition, no branch can be considered as a “basal” one. An obvious solution to this problem is the inclusion in the phylogenetic analysis of another, obviously external group, which can be used as a formal out group.

This is an important point that we missed in description of the methodology. Thank you for the opportunity to clarify it now. Indeed, in the work were initially used the unrooted maximum-likelihood reconstructions. To perform evolutionary analysis by the uniform way, we rooted all dendrograms on the most genetically distant group of E. tancrei from Mongolia and term 'basal' was not used in relation to it. The fact is that in this study we did not aim especially to find out the direction of evolution, resolving the problem needs more material both E. alaicus - E. tancrei and third species of the subgenera Ellobius, E. talpinus, which is characterized by quite complicated and significant molecular variability too (Bogdanov et al., 2015; Lebedev et al., 2020). Here we were more interested in studying genetic polymorphism in E. alaicus and E. tancrei. Furthermore, using the species Ellobius talpinus as an outgroup would reduce the resolution of phylogenetic reconstructions necessary to assess the intraspecific polymorphism of the recently diverged species E. tancrei and E. alaicus.

We added the information about rooted trees to the Material and methods and Results sections,

 see lines 165-166, ' Final images of phylogenetic trees rooted on the genetically remotest group (i.e. mole voles from Mongolia) in all cases'

and corrected capture to figures 2, 5, and 7 with " The dendrogram was rooted on the group of mole voles from Mongolia."

  1. Secondly, there is a too big gap between the Results sections, in which all the findings are presented very punctually, and the Discussion, in which the conclusions are too general. Because of this, in some places it is difficult to follow the logic of the authors. Let me explain this. So, it is said that there is a mito-nuclear discordance. The idea is clear, but, it would be useful to summarize at the beginning of the Discussion section how this discrepancy manifested itself (which individuals/populations produced different topologies when using different genes).

After rereading our manuscript in the light of this remark, we really found a great lack of auxiliary information for correlating data. We are very grateful for your fresh look, which helped us to fill this gap. We tried to change and concretize some our conclusions, spoken too abstractly, by additional information, examples, and explanations. Thus, we add a quite large text fragment clarifying “mito-nuclear discordance” (see Paragraphs 1 and 3 in Discussion).

  1. It should also be clearer explained on which basis the population (individuals) from Gulcha (locality 1) are positioned as hybrids.

Thank you for the note. The hybrid zone in the Gulcha locality was described according to karyological data in our previous paper [18], it was reflected in Paragraph 2 in Discussion too. In this study, we confirm the hybrid nature for 27491 by the karyotyping (Figure 2a) and for other specimens, which kept the E. alaicus chromosomes, by molecular markers. Interestingly, three animals (27491, 27497, and 27498) were caught from the same hole, they can be treated as closely related (mole voles are social rodents), but with different karyotypes.

  1. In the Abstract the sentence “Analysis of IRBP demonstrated the presence of the specific genotype in most of E. alaicus specimens, but also revealed typical for E. tancrei haplotype” is not clear. Maybe the authors meant “Analysis of IRBP revealed not only haplotypes specific for E. alaicus, but also haplotypes shared by E. alaicus and by E. tancrei

Thank you very much for the comment. We have edited the abstract, clarifying the wording in order to avoid double interpretation, see lines 19-26: 'Analysis of IRBP demonstrated presence of the specific genotype in most of E. alaicus specimens, but also revealed the haplotype, typical for E. tancrei, in some Alay mole voles'.

  1. The scheme (Figure 1) lacks some important points, for example the locality 2 (Taldyk pass).

Accepted. Missing points added. The legend was modified (Figure 1).

Reviewer 2 Report

The MS is really well-written. I only have a few minor revisions.

1) Line 35. Please, add the genus Talpa

2) Lines 61-71. Please, may you add your predictions?

3) Lines 74-87. Please, may you add details on sample collection?

4) Figure 6 should be improved. It is not self explanatory. Add some information at nodes.

Author Response

We are grateful for the appreciation of our work and for the comments.

  1. Line 35. Please, add the genus Talpa

The genus Talpa, of course, is of great interest from the point of view of chromosomal diversity. However, in phrase, previous to one, which provides list of chromosomally polymorphic genera, we indicated that we would like to focus in this study specifically on subterranean rodents, see Lines 37-40: 'Moreover, extensive intraspecific chromosomal variability was described for many subterranean rodents [3,4]. The most chromosomally diverse genera, such as Ctenomys, Oryzomys, Fukomys, Nannospalax, Ellobius, etc. are highly specialized; underground habitations limit their dispersal and shape foraging, mating, and breeding [5–7].'

Therefore, despite the undoubted convergent similarity of moles with underground rodents, we do not include them in this list, as we do not include the genus Sorex, widely known for its chromosomal variability too. It seems to us, speaking about the subtleties of evolutionary processes, it would not be quite correct to put such distant taxa as Rodentia and Eulipotyphla in one list.

  1. Lines 61-71. Please, may you add your predictions?

Thank you for your comment, the additions have been made:

Lines 79-82 We can suggest a more complex picture of genetic polymorphism in both species than is currently known, considering the extremely diverse landscape and climatic conditions of these mountainous regions.

  1. Lines 74-87. Please, may you add details on sample collection?

We really missed this important part of the description of the material, now all the necessary information has been added, see lines 89-90 and Figure 1:

The total number of specimens used for the molecular analyses was 60, from which 56 mole voles were karyotyped. Geographical locations of capture points of the mole voles were shown in Figure 1 We also added GenBank numbers to the Table S1.

  1. Figure 6 should be improved. It is not self explanatory. Add some information at nodes.

Thank you very much for the comment. Accepted. We agree that the drawing was not informative enough. It has been modified and executed in the same style as other phylogenetic reconstructions. We hope that now, with the help of color coding, we have made Figure 6 as clear as possible.

Round 2

Reviewer 1 Report

In my opinion, the authors adequately responded to all comments. The manuscript is ready for publication.